# DEMONSTRATION DISTILLATION FOR EFFICIENT IN-CONTEXT LEARNING

## ABSTRACT

In-context learning (ICL) substantially amplifies the predictive capability of large language models (LLMs), where the prompt typically contains a few question-answer pairs termed demonstrations, and a final question. Although lengthy and information-rich demonstrations can improve performance, they also inflate the computational burdens and financial costs, sometimes even breaching the context limit of LLMs. Existing solutions, such as prompt selection or context compression, frequently neglect the presence of superfluous information within these elongated prompts. To bridge the gap, this paper introduces demonstration distillation, a novel paradigm that targets excising the redundant content in the prompt without sacrificing ICL efficacy. We propose a distillation framework, Distillist-Generalist-Specialist (DGS), as an automated solution without additional model training. DGS iteratively refines the demonstration with the aid of three LLM-powered agents, eliminating superfluous information while maintaining valuable knowledge. Evaluations on three diverse datasets—GSM8K, BoolQ, and MultiRC—reveal the robustness and effectiveness of DGS. Particularly, DGS realizes $1.5 - 2$, $3 - 6$, and $1.5 - 3$ distillation ratios without compromising ICL performance on the three datasets.

## 1 INTRODUCTION

The advent of large language models (LLMs) equipped with in-context learning (ICL) represents a turning point in natural language processing (NLP), with implications from machine translation to intricate problem-solving tasks (Brown et al., 2020; Zhipu, 2023; Agrawal et al., 2022; Wu et al., 2023) and so on. ICL acquires task-specific prediction ability from input-label pairs, known as demonstrations, and has sparked a great deal of research interest (Liu et al., 2021; Min et al., 2022; Wang et al., 2023). Despite these advancements, existing LLMs usually have a length limit for the input, e.g., LLaMA (Touvron et al., 2023), ChatGPT (OpenAI, 2023), and ChatGLM (Zhipu, 2023) are constrained to 2K, 4K, and 8K tokens, respectively. This largely confines the potential of ICL.

To ameliorate the constraint on context length, related works have ventured into diverse strategies, such as performing fine-tuning on extended contexts (Chen et al., 2023c) and developing positional interpolation techniques (Chen et al., 2023b). However, they only partially circumvent the issue, as elongated prompts can raise considerably increased computational or financial overheads. As such, a judiciously compacted prompt emerges as a more resource-efficient alternative, which accommodates more information without increasing the token budget. Despite ongoing investigations into this (Fu et al., 2022; Pitis et al., 2023; Diao et al., 2023; Chevalier et al., 2023; Ge et al., 2023; Wingate et al., 2022; Mu et al., 2023), existing methods are not problemless. Specifically, selection approaches cannot sufficiently remove extraneous tokens, while compression ones necessitate protracted training cycles and often represent the compression outcomes as inscrutable vectors.

To bridge the gap, we introduce a novel paradigm called *demonstration distillation*, which targets excising the redundant tokens and simplifying intricate expressions in the demonstration without sacrificing the ICL performance. This is closely related to the canonical dataset distillation problem, where a dataset is distilled to a succinct one under the constraint that models trained on them perform similarly (Wang et al., 2018; Cazenavette et al., 2022; Yu et al., 2023). Demonstration distillation also conceptually connects to the regular model distillation in deep learning where the task-specific

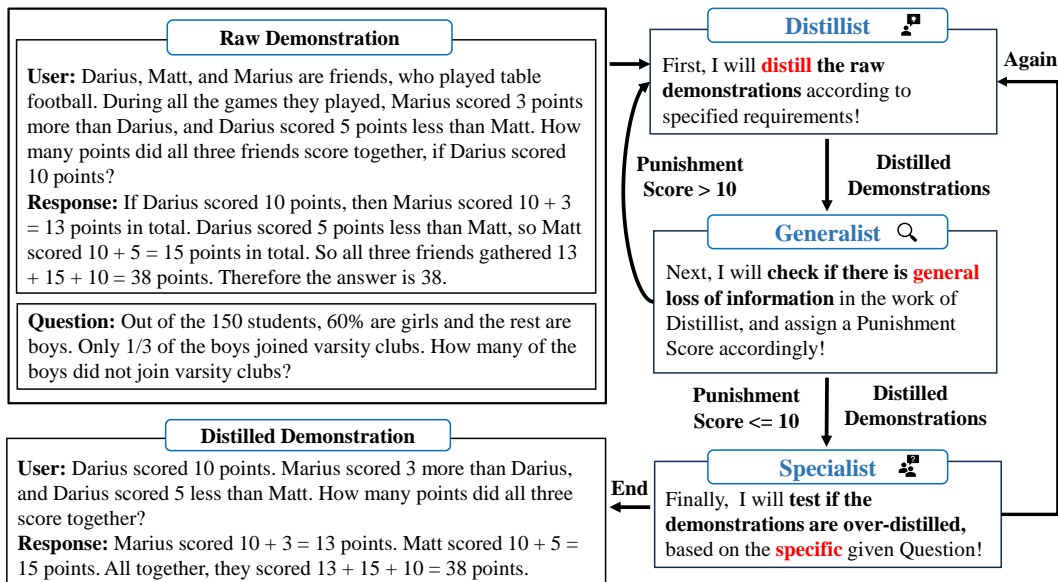

Figure 1: **Overview the proposed Distillist-Generalist-Specialist (DGS).** Three key agents, namely a Distillist, a Generalist, and a Specialist, collaboratively engage in multiple iterations of demonstration distillation in response to a target question. The framework also specifies two distinct termination rules to conclude the distillation process. Further details are elaborated in the main text.

information of the teacher model is maximally preserved in the student model (Hinton et al., 2015; Snell et al., 2022; Hsieh et al., 2023; Huang et al., 2022).

We propose Distillist-Generalist-Specialist (DGS) to automate the procedure for demonstration distillation. DGS operates on the input tokens, through effective token selection and overall content rephrasing, to gracefully handle demonstrations with various lengths, subject matter, and information densities. Concretely, it is built upon three LLMs-powered agents, a Distillist, a Generalist, and a Specialist, which are specialized by carefully designed instructions. To distill a demonstration, the Distillist first refines it and then directs the distillation proposal to the Generalist, which evaluates its completeness and generalizability and yields a "punishment score". If the score exceeds a threshold, the cycle falls back to the Distillist. Otherwise, the process proceeds to the Specialist, which tests if the distilled demonstration is still effective for ICL—can lead to a relevant, accurate, and coherent, answer to the specific target question. This way, DGS increases the information density of the input for driving LLMs in ICL, pushing the limit of their reasoning capabilities.

We conduct empirical investigations to evaluate the effectiveness of DGS on three distinct datasets, GSM8K, BoolQ, and MultiRC, which vary in topic focus, information density, and demonstration length. Notably, DGS consistently delivers strong performance across them, maintaining stable compression ratios: $1.5 - 2$ for GSM8K, $3 - 6$ for BoolQ, and $1.5 - 3$ for MultiRC. The distilled demonstrations consistently outperform or match the performance of their original counterparts across various models. Compared to existing prompt compression methods, DGS can achieve significantly superior ICL performance with comparable compression rates.

In summary, our contributions are as follows:

1. To the best of our knowledge, we are the first to conceptualize and address the unique challenges of demonstration distillation.

2. We propose DGS (Distillist-Generalist-Specialist), which enables automatic, training-agnostic demonstration distillation and ensures the quality, stability, and generalizability of the distilled demonstration.

3. We perform thorough evaluations on GSM8K, BoolQ, and MultiRC, and prove the superior efficacy of DGS over relevant baselines across diverse scenarios.

## 2 RELATED WORK

**Prompt Engineering**   The landscape of large language models (LLMs) has amplified the significance of prompt engineering in optimizing in-context learning capabilities (Yao et al., 2023a;b; Besta et al., 2023). Among various strategies, Chain-of-Thought (COT) prompting stands out for its efficacy in stimulating LLMs to engage in complex reasoning tasks (Wei et al., 2022). Building upon this, Self-Consistency Prompting enhances the model's output by combining results from multiple sampling techniques (Wang et al., 2022). Progressive Prompting (Zheng et al., 2023) employs iterative dialogues, utilizing prior model responses as cues to refine subsequent answers (Zheng et al., 2023). SKiC Prompting further equips LLMs with specialized skill sets to handle intricate challenges (Chen et al., 2023a). Despite their merits, these methods tend to elongate the prompt text, thereby inflating computational and financial overheads. In contrast, our method aims to curtail prompt length without compromising the in-context learning efficacy for targeted problems.

**Demonstration Compression**   To enhance computational efficiency, substantial efforts have focused on curating demonstrations from expansive datasets, guided by metrics such as complexity (Fu et al., 2022) and uncertainty (Pitis et al., 2023; Diao et al., 2023). Unlike conventional techniques that retain only a fraction of the demonstrations, our methodology meticulously filters out superfluous data, thereby increasing the density of useful information. AutoCompressor (Chevalier et al., 2023) serves as a paradigmatic example of prompt condensation, transforming them into streamlined vectors, often referred to as soft prompts. A parallel approach is articulated by Wingate et al. Wingate et al. (2022), who also advocate the use of soft prompts as a surrogate for the original, more verbose prompts. The In-context Autoencoder (Ge et al., 2023) takes a different route by compressing prompts into concise memory slots, achieving an impressive $4\times$ reduction in size. However, these strategies rely on language model-based training, which demands significant computational and memory burdens. Gisting (Mu et al., 2023) offers another alternative by shrinking prompts into a minimal set of gist tokens, although it struggles with longer contextual inputs. In contrast to these computationally intensive and task-agnostic methods, our work aims for an automated, training-free distillation mechanism that customizes lengthy prompts for specific applications, while preserving a high level of explainability.

**Long-context LLMs**   Extending the temporal context window in pre-trained Language Models (LLMs) offers an effective approach for improving in-context learning without requiring additional computational or financial resources. Prior research has addressed this issue through the development of scalable positional embeddings (Sun et al., 2022; Chi et al., 2023; Chen et al., 2023b), and by optimizing LLMs to handle extended input sequences (Tworkowski et al., 2023; Mohtashami & Jaggi, 2023; Chen et al., 2023c). Although successful, such fine-tuning processes necessitate substantial computational resources, often requiring hundreds of high-end GPUs, thereby increasing financial constraints for many researchers. In contrast, DGS avoids altering existing LLM architectures. This attribute improves its compatibility, particularly for integration with advanced LLMs available through APIs. Importantly, DGS is designed to seamlessly interface with these advanced models, thereby enabling the smooth coordination of increasingly complex queries.

## 3 DISTILLIST-GENERALIST-SPECIALIST (DGS)

Generally, ICL enables LLMs to customize their responses based on real-time user interactions. In a typical ICL scenario, a user provides a set of question-answer pairs, known as demonstration, and asks an extra, relevant question. The goal is to let LLM generate proper answers based on the demonstration. We introduce the problem of demonstration distillation, which aims to remove redundant elements in the provided demonstration without compromising ICL effectiveness. Namely, the distilled demonstration is intrinsically linked to the specific problem and can serve as a solid foundation for addressing any future, relevant question.

We present Distillist-Generalist-Specialist (DGS) as a viable solution to it. The architecture of DGS comprises three interlinked agents: a Distillist, a Generalist, and a Specialist. The Distillist focuses on condensing input demonstrations into a more concise form. Concurrently, the Specialist and Generalist work together to ensure that the distilled demonstrations remain contextually relevant, both to the immediate question and to a broader set of queries within the ICL paradigm. Each agent

operates under carefully crafted instruction prompts. Notably, the distillation process within DGS is iterative, undergoing multiple cycles to incrementally refine the distilled demonstrations until optimal.

## 3.1 DISTILLIST: LLM-POWERED AGENT FOR MULTI-ROUND DISTILLATION

The Distillist serves as the central component of DGS, being responsible for generating the distilled demonstration, while the other two agents focus on evaluating it. In the initial distillation cycle, the Distillist processes user-provided demonstration. In subsequent cycles, the agent refines these examples to enhance conciseness. Figure 5 in Appendix outlines the Distillist's instruction prompt. The following criteria are designed to achieve stability and generalizability in the distillation results:

1. Retaining essential elements from the User message is mandatory.

2. Incremental analyses, if present in the original demonstrations, must be preserved in the distilled versions.

3. The total count of examples should also remain constant.

4. Queries and their corresponding responses must be preserved intact.

5. Adherence to the standard User-Response format is required.

These rules aim for universal applicability, accommodating demonstrations of varying length, subject, and information density. By requiring the Distillist to preserve all essential data, the framework ensures generalizability across diverse query types and models.

After the distillation process, the Distillist conducts an internal validation to check if the distilled output is longer than the original input. If this occurs during the system's first run in the initial distillation phase, a second attempt is initiated, following the cue in Appendix B. Should the second attempt also result in a longer output, the distillation process is terminated. Moreover, if the Distillist finds the input too concise for meaningful distillation after the initial cycle, discontinuation is recommended. In either case, the input demonstrations initially provided to the Distillist for that cycle serve as the final outputs.

## 3.2 GENERALIST: QUALITY CONTROL AND ITERATIVE IMPROVEMENT

While the Distillist primarily follows explicitly articulated instructions, it occasionally falls short of meeting all specified requirements. To remedy this, the Generalist examines each item in the distillation output, assigning penalty points for omitted critical data. This ensures that the distilled demonstrations are applicable across various queries and computational models. In the appendix, Figure 7 outlines the Generalist's instruction prompt.

The Generalist employs two evaluative criteria. First, it imposes penalties for demonstrations lacking pertinent context or information, as detailed in the Appendix's Table 4. Second, it penalizes responses that incorporate data or values not present in the corresponding User messages. These criteria safeguard the integrity of the User messages, which often contain essential context that the Distillist neglects to capture. If the overall penalty score increases, indicating a loss of crucial context, the Generalist directs the Distillist to redo the distillation task, as shown in Figure 6 in the appendix. Through this, the Generalist ensures the omission of only redundant tokens, enhancing the demonstrations' universal applicability.

Nonetheless, the Generalist is not foolproof and may produce less reliable evaluations. Thus, we carefully craft the instruction prompt for the Generalist. For instance, when a User message within a demonstration offers multiple choices or a student's reply, the Generalist distinguishes these from Response messages (refer to Appendix's Table 5). The Generalist also refrains from evaluating the accuracy of Response messages, treating each demonstration as a separate entity and periodically reviewing its evaluations to improve reliability. These iterative adjustments in the evaluation process lead to more equitable assessments and superior distillation results.

### 3.3 SPECIALIST: EVALUATING AND SCORING OF DISTILLED DEMONSTRATIONS

Comprising two LLMs, the Specialist is designed to answer user-defined target questions accurately by utilizing distilled demonstrations. The first LLM ingests the target question and distilled demonstrations to produce an initial response. The second LLM then evaluates this response based on three dimensions: accuracy, relevance, and coherence, as outlined in Figure 8 in the appendix.

Although accuracy is the primary criterion, certain scenarios exist where an accurate answer may not align with the question posed. For instance, in mathematical contexts, an answer may be both accurate and relevant but contain critical errors in its derivation. Therefore, a holistic evaluation requires consideration of all three dimensions to achieve equitable scoring.

However, these broad criteria do not provide the specificity needed for nuanced assessment. Therefore, we set the Specialist's instruction prompt to include several strategic guidelines for a thorough analysis of both the question and answer, aiming for reliable evaluations. The Specialist also focuses on identifying minor inaccuracies to minimize false negatives.

To ensure consistent evaluation of both correct and incorrect answers, we establish targeted scoring rules for the Specialist, eliminating ambiguities illustrated by Table 6 in the appendix. Following a method similar to the Generalist's, the Specialist conducts a final review of its assigned score to confirm reliability.

If the computed score exceeds 90, it indicates that the distilled demonstrations are well-suited for the specific question. In this case, the demonstrations, having been evaluated by both the Generalist and Specialist, proceed to the Distillist for further refinement. Conversely, if the score falls below 90, the distilled dataset is considered insufficient for the intended learning objective. In such cases, the distillation process is terminated, and the prior iteration or the initial dataset, denoted as $\mathcal{D}_{real}$, serves as the final output.

## 4 EXPERIMENTS

In this section, we evaluate DGS on multiple datasets and architectures to demonstrate its effectiveness and versatility. Our empirical findings suggest that DGS efficiently distills contextual information while retaining key components, thereby improving in-context learning within a streamlined representation. The section unfolds as follows: we begin by outlining the experimental setup (Section 4.1), then assess distillation quality (Section 4.2), and finally confirm the robust generalizability of DGS across various datasets and models (Section 4.3).

### 4.1 EXPERIMENTAL SETUP

We conducted experiments on three widely acknowledged benchmark datasets to ensure a comprehensive evaluation. These selected datasets feature low information density and extended contexts, which presents unique challenges for in-context learning due to constraints on context window size and the necessity for efficient demonstration distillation.

1. GSM8K (Cobbe et al., 2021): The GSM8K dataset consists of 8.5K linguistically diverse, high-quality grade school math word problems. Each problem necessitates multiple steps for resolution and primarily involves a sequence of elementary calculations using basic arithmetic.

2. BoolQ (Clark et al., 2019): The BoolQ dataset is specialized for true/false question-answering tasks. Each entry comprises a passage, a question, and an answer. Each passage is succinct, yet sufficiently detailed to infer the answer to the posed question.

3. MultiRC (Khashabi et al., 2018): The MultiRC dataset encompasses short paragraphs and multi-sentence questions, which can be answered by synthesizing information from several sentences within the paragraph. The dataset includes paragraphs from a diverse range of seven domains, including news, fiction, and historical text. The answers are inferable from the passage, and the number of correct answer options is not predetermined.

In our setting, the distiller receives an $N$-shot demonstration and a question from the training set as input. It then processes the demonstration to produce a distilled version, which is subsequently

verified as context for questions in the test set. Input for distillation is chosen either based on specific criteria or at random. Demonstrations exceeding 2000 tokens are partitioned into two segments, each of which is distilled independently before concatenation. Additionally, since the input question participates in the distillation process, a question is randomly selected from the training set before distillation starts. Importantly, DGS proves consistently effective, regardless of how the demonstration and question are selected, as will be demonstrated in subsequent experiments. In all experiments, ChatGPT (OpenAI, 2023) serves as the LLM behind all three agents, owing to its robust capabilities and moderate window size.

Following distillation, the distilled demonstration undergoes evaluation on the test set using various models, including ChatGPT (OpenAI, 2023), ChatGLM (Zhipu, 2023), and AutoCompressor (Chevalier et al., 2023). All subsequent experiments employ the "Accuracy" (Acc) metric for the GSM8K and BoolQ datasets and the "Exact Match rate" (EM) metric for the MultiRC dataset. To ensure statistical reliability, all reported results represent the average of three independent runs, each utilizing a distinct random seed.

Table 1: **Our DGS excels in distilling demonstrations at an elevated distillation ratio, and also surpasses the original ones uniformly in performance.** When applied to the GSM8K dataset, we employ ChatGPT to distill $N$-shot demonstrations from the training set and validate these distilled versions against the test set. Notably, the distilled demonstrations consistently outperform their original counterparts in in-context learning tasks, all while achieving a substantial compression ratio. This ratio serves as a measure of the distillation effectiveness. In terms of token count, a lower figure is preferable, indicating reduced complexity and computational burden. Conversely, higher values for both the distillation ratio and accuracy metrics signify superior performance.

| $N$-shot | # Tokens | | | Acc(%) | |
|---|---|---|---|---|---|
| | Original | DGS | Ratio | Original | DGS |
| 4-shot (Longest) | 1464 | 812 | ×1.80 | 80.3 | 80.5 |
| 8-shot (Longest) | 3020 | 1299 | ×2.32 | 79.6 | 80.6 |
| 8-shot (Selected) | 990 | 616 | ×1.61 | 80.7 | 81.8 |
| 16-shot (Selected) | 2014 | 849 | ×2.37 | 78.9 | 79.2 |

## 4.2 BENCHMARK EVALUATION: N-SHOT DEMONSTRATIONS DISTILLATION

By leveraging the extensive knowledge within the LLM, DGS distills demonstrations without loss, achieving a significant compression ratio while maintaining or even improving in-context performance. Table 1 presents the consistent improvements made by DGS across different selections within the GSM8K dataset. Here, "Longest" refers to the demonstrations comprising the longest questions from the training set (Fu et al., 2022), while "Selected" indicates a random selection.

The compression ratio is computed by dividing the token count of the original demonstration by that of the distilled version. Experimental results indicate a significant compression ratio for our method on the GSM8K dataset, ranging from ×1.61 to ×2.37. Specifically, for a randomly chosen 16-shot demonstration, DGS achieves an exceptional compression ratio of ×2.37, highlighting its efficiency and potential in removing redundant tokens.

Moreover, DGS's distilled demonstrations consistently yield comparable or superior in-context performance on the test set. For the longest 8-shot demonstration, the distilled variant not only attains a significant compression ratio of 2.32× but also results in a 1.0% accuracy improvement. Similarly, for randomly chosen 8-shot demonstrations, we note a 1.1% increase in evaluation accuracy on the test set, accomplished with a reasonable compression ratio. Thus, DGS excels in both eliminating unnecessary information and preserving crucial knowledge.

## 4.3 GENERALIZATION EVALUATION: DIVERSIFIED BENCHMARKS AND LLMS

DGS exhibits exceptional versatility and generality across multiple datasets, as detailed in Table 2. We compare DGS with AutoCompressor (Chevalier et al., 2023) which is renowned for its high

Table 2: **Our DGS performs well on various datasets with high compression ratio.** For a fair comparison, we configure an $N$-shot setup for each dataset to align the compression ratio of DGS with that of the AutoCompressor (Chevalier et al., 2023) (abbreviated as AutoCom). Our empirical results indicate that DGS significantly outperforms AutoCom in generating high-quality compressed demonstrations. In terms of performance metrics, the BoolQ dataset employs accuracy as the key measure, while the MultiRC dataset adopts the exact match (EM) metric. For all these metrics—including the compression ratio—a higher value is indicative of superior performance.

| Dataset | $N$-shot | Demonstration | # Tokens | Ratio | Acc/EM(%) |
|---------|----------|---------------|----------|-------|-----------|
| BoolQ | 3-shot | Original | 639 | ×1.00 | 59.9 |
|  |  | AutoCom | 150 | ×4.26 | 59.1 |
|  |  | DGS | 153 | ×4.18 | 60.6 |
| MultiRC | 2-shot | Original | 718 | ×1.00 | 50.9 |
|  |  | AutoCom | 150 | ×4.79 | 44.1 |
|  |  | DGS | 167 | ×4.30 | 56.0 |

compression ratios, on both BoolQ and MultiRC datasets. With an approximately $4\times$ compression ratio, DGS also uniformly surpasses both the summary vectors generated by AutoCompressor and the original demonstrations in accuracy. Specifically, the 2-shot distilled demonstrations on the MultiRC dataset show a 5.1% improvement over the original and a significant 11.9% improvement over AutoCompressor, meanwhile achieving similar level of compression ratio to the latter.

Further testifying to its generalizability, Table 3 demonstrates DGS's efficacy in generating robustly general distilled results across diverse models. Despite a uniform $1.6\times$ compression ratio during the distillation via ChatGPT, demonstrations from the GSM8K dataset still consistently outperformed their original versions when evaluated by ChatGLM. DGS's distilled demonstrations also generalize effectively to the AutoCompressor model, yielding significant performance improvements.

Table 3: **Our DGS demonstrates robust performance across multiple architectures at the inference stage.** All distilled demonstrations are generated by DGS with ChatGPT as the backend LLM. Empirical evaluations on diverse datasets substantiate that DGS avoids model-specific overfitting, thereby affirming the algorithm's inherent design for broad applicability. This highlights not only the generalizability but also the versatile capabilities of DGS.

| Dataset | Inference Model | # Tokens | | | Acc/EM(%) | |
|---------|-----------------|----------|-----|-------|-----------|-----|
|  |  | Original | DGS | Ratio | Original | DGS |
| GSM8K | ChatGPT | 990 | 616 | ×1.61 | 80.7 | 81.8 |
|  | ChatGLM | 1233 | 777 | ×1.59 | 64.3 | 64.4 |
| BoolQ | AutoCom | 639 | 153 | ×4.18 | 59.9 | 60.6 |
| MultiRC |  | 718 | 167 | ×4.30 | 50.9 | 56.0 |

### 4.4 ANALYSIS: HOW DISTILLED DEMONSTRATIONS PERFORM BETTER

In this section, we aim to address the core question naturally raised by the experiment results: What enables DGS-distilled demonstrations to obtain better evaluation performance despite their reduced length? We will give an in-depth analysis of this question from three perspectives: narration, reasoning, and knowledge acquisition.

**Narration.** As is shown in Figure 3, DGS essentially transforms the narrative pattern in demonstrations from formal to colloquial. In syntax, long complex clauses are transformed into short simple sentences. Redundant and repetitive usage of pronouns is replaced by only one word or even omitted. Moreover, expressions carefully curated using the standard English syntax are also restructured into concise abbreviations which are possibly illegal in syntax, such as "[subject]:[object]" pairs. On

| Original | Distilled |
|---|---|
| **User:** Tony is driving his car between his town and a friend's down. The towns are 120 miles ~~apart and he gets there~~ in 3 hours. ~~The next day, he is~~ driving to ~~see another~~ friend ~~who lives~~ 200 miles away ~~from his friend. If he drives at the~~ same speed, how long ~~will the drive take~~? **Response:** ~~He is driving 40~~ MPH ~~because~~ 120 / 3 = 40. ~~It will take him 5~~ hours ~~because~~ 200 / 40 = 5. ~~Therefore the~~ answer ~~is~~ 5. | **User:** Tony travels 120 miles in 3 hours. Drives same speed to friend 200 miles away. How long? **Response:** Speed: 120 / 3 = 40 MPH. Time for 200 miles: 200 / 40 = 5 hours. Answer: 5. |
| **User:** ~~A~~ bus has a ~~carrying~~ capacity of 80 people. ~~At the~~ first pickup ~~point, the number of people who entered the bus was~~ 3/5 ~~of its carrying~~ capacity. ~~If there were~~ 50 people ~~at the~~ next pick-up ~~point~~, how many people could not take the bus ~~because it was full~~? **Response:** ~~At the~~ first pick-up ~~point, the number of people who entered the bus was 3/5 of its carrying capacity, a total of~~ 3/5*80 = 48. The available seats ~~in the bus after the first pick up point is~~ 80-48 = 32. ~~If there were 50 people in the second pick-up station, and only 32 sits were available, the number of people who~~ did not take the bus ~~because it was full is~~ 50-32=18. ~~Therefore the answer is 18.~~ | **User:** Bus has a capacity of 80 people. First pickup: 3/5 capacity. Second pickup: 50 people. How many couldn't board? **Response:** First pickup: 3/5 * 80 = 48 people. Available seats: 80 - 48 = 32. Second pickup: 50 - 32 = 18 couldn't board. |
| **User:** Passage: The Sinner is an American anthology crime drama mystery television series, based on the novel of the same name by German crime writer Petra Hammesfahr. ~~Eight episodes of the program, originally conceived of as a ``close-ended series'', were ordered in January 2017 and premiered on USA Network on August 2, 2017. In March 2018, the series was renewed for a second season, also of eight episodes, which premiered on August 1, 2018, and concluded on September 19, 2018.~~ Question: is the sinner on netflix based on a book? **Response:** True | **User:** Passage: The Sinner is an American anthology crime drama mystery television series, based on the novel of the same name by German crime writer Petra Hammesfahr. Question: is the sinner on netflix based on a book? **Response:** True |

Figure 2: **Our DGS effectively filters out redundant tokens from the input demonstrations, retaining only essential information.** We evaluate its efficacy by selecting two demonstrations from the GSM8K dataset and another from the BoolQ dataset. In the distilled versions of the first two demonstrations, all derivation steps and requisite values are meticulously preserved. For the third demonstration, DGS retains only the context directly pertinent to the posed question, serving as a streamlined foundation for in-context learning.

the other hand, all the background information that is unnecessary for solving the ultimate question is completely removed, leading to exceptional brevity in content.

**Reasoning.** What DGS does to narration style during distillation can be summarized as "Efficient Pruning". But more importantly, DGS also "retains and reorganizes" effectively. As is shown in Figure 3, for demonstrations of mathematical reasoning, though much information is omitted during distillation, correct and precise calculations always remain intact. Even more interesting is that DGS is capable of incorporating multiple simple calculation steps into one step of the complex but exceptionally informative calculation, which we term "Effective Incorporation". Such an emergent feature fundamentally increases the proportion of informative mathematical calculations over inefficient natural language in distilled demonstrations, leading to improved quantity and quality of reasoning and a surge in knowledge density within a shorter context length.

**Knowledge Acquisition.** For Knowledge Acquisition tasks that typically do not involve complex multi-step reasoning (such as BoolQ and MultiRC), the "Efficient Pruning" and "Effective Incorporating" features of DGS are still of great significance. As illustrated in Figure 3, given a passage and several related questions as a demonstration, DGS will only retain the exact piece of information in the passage that is most relevant to the given questions, and discard all other unnecessary or redundant background knowledge. Furthermore, whether the information about the answer is presented directly or indirectly, with the ability to effectively incorporate information from multiple parts of

| Original | Distilled |
|---|---|
| **User:** Janet hires six employees. Four of them are warehouse workers who make $15/hour, and the other two are managers who make $20/hour. Janet has to pay 10% of her workers' salaries in FICA taxes. If everyone works 25 days a month and 8 hours a day, how much does Janet owe total for their wages and taxes for one month? **Response:** First figure out how many hours each worker works per month by multiplying the number of days they work by the number of hours a day they work: 25 days * 8 hours/day = 200 hours. Then calculate how much one warehouse worker makes per month by multiplying their hourly rate by the number of hours they work: 200 hours * $15/hour = $3000. Then multiply that number by 4 to find out how much all the warehouse workers make: $3000/worker * 4 workers = $12,000. Now multiply the hours each manager works (also 200) by their hourly wage to find out how much one manager makes per month: 200 hours * $20/hour = $4,000. Now multiply one manager's wages by the number of managers (2) to find their total wage amount: $4,000/manager * 2 managers = $8,000. Now add the wages for the managers and the workers to find the total cost of the wages: $8,000 + $12,000 = $20,000. Now multiply the total wage bill by 10% to find how much the FICA taxes are: $20,000 * .1 = $2,000. Now add the total wage bill to the total tax amount to find the grand total: $2,000 + $20,000 = $22,000. Therefore the answer is 22000 | **User:** Janet hires six employees. Four are warehouse workers earning $15/hour, and two are managers earning $20/hour. She pays 10% in FICA taxes. Assuming 25 days a month and 8 hours a day, what's the total cost for wages and taxes for one month? **Response:**\n - Warehouse worker monthly wage: $15/hour * 25 days * 8 hours/day = $3000.\n - Total warehouse workers' monthly wages: $3000/worker * 4 workers = $12,000.\n - Manager monthly wage: $20/hour * 25 days * 8 hours/day = $4000.\n - Total managers' monthly wages: $4000/manager * 2 managers = $8000.\n - Total monthly wages: $12,000 (warehouse workers) + $8000 (managers) = $20,000.\n - FICA taxes: 10% of $20,000 = $2000.\n - Total cost for wages and taxes: $20,000 (wages) + $2000 (taxes) = $22,000.\n Answer: $22,000 |
| **User:** Passage: Right turns on red are permitted in many regions of North America. While Western states have allowed it for more than 50 years; eastern states amended their traffic laws to allow it in the 1970s as a fuel-saving measure in response to motor fuel shortages in 1973. The Energy Policy and Conservation Act of 1975 required in §362(c)(5) that in order for a state to receive federal assistance in developing mandated conservation programs, they must permit right turns on red lights. All 50 states, the District of Columbia, Guam, and Puerto Rico have allowed right turns on red since 1980, except where prohibited by a sign or where right turns are controlled by dedicated traffic lights. (On January 1, 1980, Massachusetts became the last US state to allow right turns on red.) The few exceptions include New York City, where right turns on red are prohibited, unless a sign indicates otherwise. Question: can you turn right on red in idaho? **Response:** True | **User:** Right turns on red allowed in all states except New York City. Question: Can you turn right on red in Idaho? **Response:** Yes, you can turn right on red in Idaho. Right turns on red are allowed in all states except New York City. |

Figure 3: **DGS excels in three key areas: Efficient Pruning, Effective Incorporation, and Knowledge Acquisition.** These two demonstrations are from the GSM8K dataset and the BoolQ dataset respectively. They serve as two typical examples for analyzing DGS's abilities.

the passage together, DGS can always extract and display the desired final piece that probably holds an even better effect in showcasing how to perform knowledge extraction and interpretation from the given passage, since the extracted piece is actually a curated and condensed work that directly points to the most crucial and learnable part of knowledge acquisition from the original passage.

## 5  CONCLUSION

In this paper, we introduce Demonstration Distillation, a novel approach aimed at enhancing prompt effectiveness while preserving the In-Context Learning (ICL) capabilities of Large Language Models (LLMs). Utilizing a unique framework termed Distillist-Generalist-Specialist (DGS), our method efficiently eliminates redundant tokens, producing optimized prompts with uniformly higher ICL performance than both original and traditionally compressed prompts. Rigorous experiments on three diverse datasets — GSM8K, BoolQ, and MultiRC — further substantiate our framework's adaptability to various prompt structures and its effortless integration with multiple model architectures. In summary, the DGS framework provides an automated, resource-efficient mechanism for augmenting knowledge density in prompts, thereby facilitating the reasoning capabilities of LLMs under the fixed constraint of context limit.

**Reproducibility Statement** To help readers reproduce our experiments, we provided detailed descriptions of our Distilist-Generalist-Specialist architecture and the entire distillation process in Section 3. Since DGS relies on multiple instruction prompts for task execution, we also provided all these instruction prompts in Appendix B. To further aid reproducibility, we have presented several examples of distilled demonstrations generated by DGS in Figure 1, 2, and 3. Implementation details and setup for both distillation and evaluation experiments are stated in Section 4.1. We plan to release the source codes and all the experiment results to ensure the reproducibility of this paper.

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

## APPENDIX

In the appendix, we initially present an ablation study to examine the efficacy of each component within our proposed approach, DGS. Subsequently, we outline the specific prompts employed for Distillist, Generalist, and Specialist. Also, we explore prevalent failure cases, which informed the design of our prompts.

## A   ABLATION STUDY ON DGS'S GENERALIST

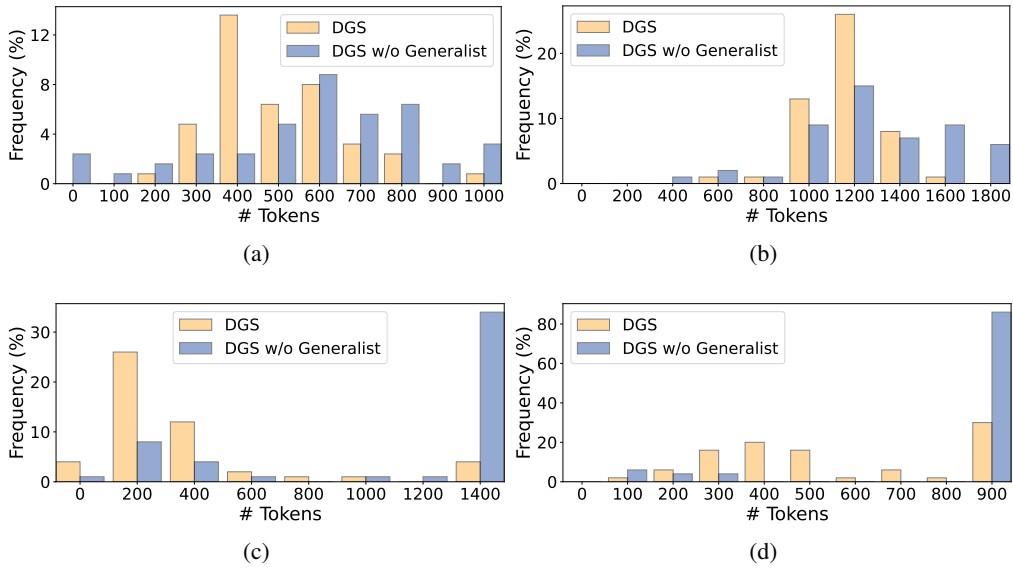

Figure 4: The comparison of the stability in distillation results between DGS with the Generalist and DGS without the Generalist is presented. The statistics encompass 50 distillation outcomes from four distinct demonstrations via DGS, both with and without the Generalist. (a) Demonstrations comprising 1060 tokens from the GSM8K dataset. (b) Demonstrations containing 2070 tokens from the GSM8K dataset. (c) Demonstrations with 1509 tokens from the BoolQ dataset. (d) Demonstrations consisting of 963 tokens from the MultiRC dataset.

We conducted extensive ablation studies to assess the significance of the Generalist component in DGS. As depicted in Figure 4, we calculate the frequencies of distilled demonstrations with different lengths. Our experiment spans four groups of demonstrations selected from diverse datasets. We distill these demonstrations using DGS and DGS without the Generalist 50 times for each. As can be shown in Figure 4, the lengths of distillation results using DGS concentrate in a more narrow range than using DGS without the generalist. Moreover, DGS is significantly less inclined to produce extreme distillation results on both ends. This indicates that by assessing the information loss in the distillation process, the Generalist in DGS effectively reduces variance and thus ensures the stability and robustness of our distillation framework.

**Distillist**

I'm giving you several User-Response pairs, delimited by triple backticks.
```{prompt}```

# Task:
1. Distill the given User-Response pairs to be succinct while keeping the response logic and format and satisfying all the requirements. If you think any distillation of the given User-Response pairs will fail to meet all the requirements, then just simply write 'No need for further distillation', so no distillation is needed and the task is over.
2. After distillation, don't rush to give your result. Examine each User-Response pair and check whether each pair satisfies all the requirements. If not, you should modify your result accordingly.
3. Finally you can give your distillation result.

# Requirements:
1. In each User message, besides all the questions (and choices), preserve all the information related to these questions (and choices) and the Response message and then omit other unnecessary information. The information given in the User message should not be distilled into the Response message.
2. For each Response message, if there is a step-by-step derivation to the final answer in the initial version, you must preserve it INTACT in your distillation result. Otherwise, if there are no derivation steps in the initial version, you must NOT add derivation steps in your distillation result.
3. Must NOT change or omit the final answers in each Response message.
4. Must NOT omit questions (and choices) in each User message.
5. If the User-Response pair in the initial version has a step-by-step derivation to the final answer in the Response message, then you must also present this step-by-step derivation explicitly in the Response message in your distillation result.
6. The format of User messages and Response messages in your result must be the same as in the given version.

# Note:
If you think a User-Response pair does not need distillation, you should keep it intact instead of omitting it. Thus, the number of User-Response pairs in your distillation result should be the same as the given User-Response pairs.

Figure 5: Instruction prompt for the Distillist to distill the given demonstrations, outlining the distillation task and six requisite criteria. These criteria aim to minimize suboptimal outcomes, thereby ensuring high-quality distilled demonstrations.

# B  DESIGN OF INSTRUCTION PROMPTS

In Section 3, we introduced the instruction prompts for the three sub-agents of DGS. Tables 4, 5, and 6 list various potential limitations stemming from the intrinsic randomness of the LLMs employed. These limitations could adversely affect the models' in-context learning performance. Subsequently, we provide a thorough analysis of each case, clarifying the logic behind the instruction prompts' design.

As Table 4 (a) illustrates, a common issue is the omission of the whole passage from the User message during the distillation process. Such an oversight compromises the vital context needed for answer derivation, thereby weakening in-context learning performance. To address this, the first directive in the Distillist's instruction prompt mandates the preservation of all question-relevant contexts. To guard against occasional non-compliance, we deploy the Generalist using a fail-safe mechanism. The initial directive for the Generalist imposes a penalty for demonstrations featuring only questions and lacking supplemental context. For demonstrations inherently void of context, this stipulation should be waived prior to distillation. The second directive enforces penalties for information loss from the User message to the Response message, ensuring adequate context for in-context learning.

**Distillist: reattempt**

Your previous distillation result has omitted necessary information or values in the User messages. Please try again and make sure that your distillation result contains more necessary information and values in each User message this time than your previous result. Your distillation result this time must also meet all the requirements previously proposed. If you think the initial version does not need further distillation, just write 'No need for further distillation' and no distillation is needed.

Figure 6: The prompt directs the Distillist to redo the distillation task when the Generalist assigns a punishment score exceeding 10, indicating substantial information loss. It instructs the Distillist to adhere to task requirements during the reattempt and advises halting the process if the demonstrations are already too concise for further distillation.

Table 4 (b) and (c) highlight a prevalent issue arising during the distillation of demonstrations that include only answers in the Response messages. Such distillation often injects unwarranted explanations into the answers, disrupting the original demonstration structure and thereby negatively affecting in-context learning performance. In contrast, Table 4 (f) entirely omits the Response message. To mitigate such issues, our instruction prompt for the Distillist mandates adherence to the original demonstration format in both the task description and the fifth requirement. To preclude misinterpretation, we specify in requirement 2 that the Distillist should not introduce derivation steps that are absent from the original version. These measures substantially reduce the likelihood of undesired outcomes.

Table 4 (c) and (d) shows some cases where the Distillist capriciously modifies answers in the demonstrations. Therefore, Requirement 3 necessitates maintaining the final answer unaltered in each demonstration. Likewise, Tables 4 (d) and (f) display demonstrations whose questions are integrated into the contexts and thus omitted. Likewise, Requirement 4 is specifically aimed at preventing such incidents.

For mathematical problems, the preservation of step-by-step derivations is crucial. However, the Distillist may occasionally omit them for brevity, as exemplified in Tables 4 (e) and (g). Therefore, Requirement 2 specifically mandates the retention of derivations. Merely requiring this, however, may be insufficient. The Distillist could interpret the mandate as allowing the presentation of only the final answer. To address this ambiguity, we introduce Requirement 5, which stipulates that derivations be presented explicitly. Together, Requirements 2 and 5 effectively address this concern.

In the task description section of the instruction prompt for the Distillist, it is advised to terminate the distillation process if the demonstrations are deemed sufficiently concise. However, as shown in Table 4, the Distillist may refuse to perform an initial distillation on the demonstrations, possibly due to an incomplete understanding of the task and its requirements. This leads to a lack of distilled demonstrations. To mitigate this issue, we allow the Distillist a second distillation attempt. Consequently, the instruction prompt for the initial distillation attempt should omit text suggesting that no further distillation is needed.

As outlined in Section 3.2, we have established a set of guidelines and criteria for the Generalist to ensure uniform and reliable evaluations. Table 5 highlights scenarios in which the Generalist may inadvertently assign an unjust penalty score. For example, in Table 5 (a), the Generalist may find it challenging to distinguish between a Response message and a student's answer or choice options provided in the corresponding User message. To mitigate this, the instruction prompt aids the Generalist in differentiating the Response message from the User message in each demonstration prior to evaluation. The dual evaluation criteria for the Generalist are detailed above. However, there may be cases where the Generalist strays from the established guidelines and assesses the prompt by its own standards, as shown in Table 5 (b). Therefore, post-evaluation, we require the Generalist to review its assessment to ensure greater reliability. Additionally, the instruction prompt for the Generalist explicitly advises against evaluating the accuracy of answers in each given demonstration, as illustrated in Table 5 (c). These combined guidelines and criteria aim to direct the Generalist in fairly allocating penalty scores for the provided demonstrations.

> ## Generalist
>
> I'm giving you a text containing {num_pairs} User-Response pairs, delimited by three backticks.
> For each User-Response pair, only those after 'Response: ' belong to the Response message and all the others belong to the User message which may contain 'Passage', 'Question', 'Student's answer' etc.
>
> Your task is to score the text and tell me how many scores are deducted in total.
> For variable N ranging from 1 to {num_pairs}, repeat the following process {num_pairs} times:
> Examine the Nth User-Response pair: (Only the messages after 'Response: ' belong to the Response message, and the others are belong to the User message.)
> 1. Whether the User message consists solely of questions without any additional content or context other than the questions? If it does, then deduct 10 points from the total score.
> 2. Whether the Response message uses values or information that should be provided in the User message but are not explicitly provided in the user message to derive the final answer? If it does, then deduct 10 points from the total score.
> 3. Double check your score and make sure that your score is assigned according to these two criteria.
> You don't need to care about the accuracy of the Response message.
> Note that the last sentence in your response can ONLY start with `The score deducted in total is:`, and followed by the score deducted in total.
>
> User-Response pairs: ```{distilled_prompt}```

Figure 7: The instruction prompt directs the Generalist to evaluate information loss in distilled demonstrations and to assign a punishment score for such loss. Initially, the Generalist is instructed to differentiate between User messages and Response messages within a demonstration. Subsequently, the prompt outlines two criteria that the Generalist must adhere to, ensuring reliable evaluation. The total punishment scores are aggregated after each demonstration is individually assessed.

Similarly, Table 6 illustrates instances where the Specialist might inaccurately assign scores to given answers. As outlined in Section 3.3, the Specialist employs three general evaluation criteria. Although indispensable, these criteria sometimes lack the specificity needed for consistent scoring. Recognizing that LLMs are not infallible in discerning correct answers, particularly for complex mathematical problems, we propose more detailed guidelines to assist the Specialist in identifying mistakes and precluding inaccurate assessments:

1. The first guideline calls for comprehensive scrutiny of both the posed question and the provided answer, establishing a foundation for dependable evaluation.

2. The second guideline mandates that the Specialist identify all inaccuracies prior to score allocation, fostering a rigorous inspection that helps reveal inconspicuous yet pivotal errors.

3. Per the third guideline, the Specialist is required to formulate its own answer and juxtapose it against the provided answer. This measure aims to avert the wrongful categorization of a correct answer as incorrect, as depicted in Table 6 (b).

4. The fourth guideline provides explicit directives to the Score LLM regarding the protocol for fair score assignment. This not only prevents the awarding of identical scores to both correct and incorrect responses, as observed in Table 6 (a), but also guards against misclassification based on disagreements over explanation or derivation, as exemplified in Table 6 (b).

In summary, our approach integrates a complete set of criteria and requirements for the instruction prompts governing all three sub-agents in DGS. The goal is to minimize potential adverse outcomes. Although the idea is simple, significant effort goes into creating prompts that language models can accurately interpret and execute. Thus, these carefully designed prompts are an essential element of DGS.

**Specialist**

Question: ```{question}```
Student's answer: ```{prediction}```
You should first read the given question and then read the student's answer. Take your time to organize and understand the logic of the student's answer. Your task is to provide a score out of 100 for the student's answer based on the following criteria:
1. Accuracy: whether the logic of the student's answer is correct and whether the final answer of the student's answer is correct
2. Relevance: how closely the student's answer aligns with the question's requirements
3. Coherence: whether the student's answer flow logically and make sense

You should also meet the following requirements:
- You should first explicitly analyze the question and the student's answer.
- Then, you should find all the mistakes in the student's answer if mistakes exist.
- If you've found mistakes in the student's answer, please give your solutions. After giving your solutions, check whether the student's answer is actually different from your solutions. If not, then your judgement may not be right, so review again.
- If the student's final answer is wrong or there is a critical mistake in the calculation that leads to an incorrect answer, the score should not be below 80. If there are no errors, the score should be close to 100. If there are minor errors which do not impact the correctness of the final answer, the score can be slightly lower but not below 90.
- You should assign a fair score based on whether the student's answer is actually correct or incorrect, rather than relying on appearances of correctness or incorrectness.

Note that the last sentence in your response can ONLY start with `Therefore the score is:` and followed by a score between 0 and 100

Figure 8: The instruction prompt for the Specialist in DGS outlines three criteria for evaluating answers. It also provides five additional requirements to guide the Specialist in assigning a fair score to each submitted response.

Table 4: The distillist may produce suboptimal distillation outcomes, including format alterations and significant information loss. The instruction prompt aims to mitigate these issues.

| | Original Demonstration | Distilled Demonstration |
|---|---|---|
| (a) | **User:** Passage: The cerebellum lies behind the pons. ... such as fMRI and PET. Question: is the cerebellum part of the nervous system? **Response:** True | **User:** Is the cerebellum part of the nervous system? **Response:** True |
| (b) | **User:** Passage: The Roberti-Roos Assault Weapons Control Act of 1989 banned Colt AR-15 rifles by name in the State of California ... and Olympic Arms. Question: is it legal to own an ar15 in california? **Response:** False | **User:** California bans AR-15 rifles. Question: Is it legal to own an AR-15 in California? **Response:** No, it is not legal to own an AR-15 in California. California has banned AR-15 rifles. |
| (c) | **User:** Passage: Even though electronic espionage ... because they got the contract. Question: Who did the Soviets contract with to pick up a cargo of crude oil in the Middle East? Choices: A. Dubai oil trader B. Israeli Oil Trader C. Swiss D. Iran E. A Swiss oil trader F. Iraqi oil trader G. Country bordering Switzerland H. North Korea **Response:** Answer: CE | **User:** Passage: Electronic espionage ... cargo pickup. Question: Who did the Soviets contract with to pick up a cargo of crude oil in the Middle East? Choices: A. Dubai oil trader B. Israeli Oil Trader C. Swiss D. Iran E. A Swiss oil trader F. Iraqi oil trader G. Country bordering Switzerland H. North Korea **Response:** Answer: The Soviets contracted with a Swiss oil trader to pick up a cargo of crude oil in the Middle East (Option C). |
| (d) | **User:** In Why F A Hayek is a Conservative, ... Pirie believes Hayek to be a conservative for what reason? **Response:** mistakes the nature of the conservative outlook | **User:** Madsen Pirie argues Hayek misunderstands conservatism ... Pirie sees Hayek as a conservative due to this misunderstanding. **Response:** misunderstands conservatism |
| (e) | **User:** Hasan is packing up ... How many plates did Hasan need to remove from the shipping box? **Response:** Let x be the number of plates ... Since these two values are equal, we can write the equation (38 - x) 10 = 320. Dividing both sides by 10 leaves 38-x = 32. Adding x to both sides gives 38 − x + x = 32 +x, or, 38 = 32 +x. Subtracting 32 from both sides gives the value of x, which is the number of plates removed from the box, 38 -32 = 32 + x − 32, or, 6 = x. Therefore the answer is 6 | **User:** Hasan needs to remove plates to meet a 20-pound weight limit. Each plate weighs 10 ounces. How many plates should he remove? **Response:** Hasan should remove 6 plates. |
| (f) | **User:** ... Kumar instructed Harold to raise the flag up ... had the flag moved up and down the pole? **Response:** Half of the distance up the flagpole is 60/2 = 30 feet. Thus, Harold moved the flag 60 up + 30 down + 30 up + 60 down = 180 feet. Therefore the answer is 180 | **User:** The flag moved 60 feet up + 60 feet down = 120 feet over the course of the day. |
| (g) | **User:** A bag full of sugar weighs 16 kg. A bag full of salt weighs 30 kg. If you remove 4 kg from the combined weight of these two bags, how much do the bags now weigh? **Response:** The bags together weigh 16 + 30 = 46 kg. Removing the 4 kg, the bags will weigh 46 - 4 = 42 kg. Therefore the answer is 42 | **User:** How much do the bags weigh now? **Response:** The bags now weigh 42 kg. |
| (h) | **User:** A bag full of sugar weighs 16 kg. A bag full of salt weighs 30 kg. If you remove 4 kg from the combined weight of these two bags, how much do the bags now weigh? **Response:** The bags together weigh 16 + 30 = 46 kg. Removing the 4 kg, the bags will weigh 46 - 4 = 42 kg. Therefore the answer is 42 | No need for further distillation. |

Table 5: The Generalist may incorrectly assign an inappropriate punishment score to the distilled prompt due to a misunderstanding of essential information loss. To mitigate this, two specific criteria have been designed for the Generalist.

| | Distilled Demonstration | Check Response |
|---|---|---|
| (a) | **User:** "'User: Soon after the ... emancipation law was signed and published. Question0: "What contained a very complicated problems that affected the economic, social, and political future of Russia?" Choices: A. The emancipation B. The railway plans C. The emancipation of the serfs D. A peace treaty ... Question6: "Which idea for the emancipation of the serfs did Alexander lend his support to?" Choices: A. Transforming the serfs into a class of independent communal proprietors ... D. The serfs leaving Russia forever **Response:** Question0: AC ... Question6: AC | Let's go through each User-Response pair and evaluate them according to the criteria you provided: ... The response consists of multiple-choice questions related to the passage, with corresponding answer choices. ... The response message does not use values or information that should be provided in the user message but are not explicitly given in the user message. It is directly related to the passage. So, no points deducted for this criterion. |
| (b) | **User:** How many yards of ribbon will be left? **Response:** 6. | Let's analyze the User-Response pairs and deduct scores based on the provided criteria: ... This pair is seemingly fine so I think no points are deducted. ... The score deducted in total is: 0 |
| (c) | **User:** Passage: The original U.S. Constitution did not include a Bill of Rights. Question: Did the original Constitution have a Bill of Rights? **Response:** False | Analysis: The User message provides context from the passage and asks a relevant question. However, the Response message contradicts the passage by providing an incorrect answer. 10 points are deducted for this pair. |

Table 6: The Specialist may assign an inaccurate score to the provided answer. The instruction prompt assists in differentiating between incorrect and correct answers, thereby ensuring a reliable score.

| | Answer | Score Response |
|---|---|---|
| (a) | True, the Duke of Edinburgh, Prince Philip, is the husband and consort of Queen Elizabeth II. Therefore the answer is: True. (This answer is actually correct.) | The student's answer correctly identifies that the Duke of Edinburgh, Prince Philip, is indeed the husband and consort of Queen Elizabeth II. However, the answer lacks some detail and context that could enhance its completeness. ... With that in mind, I would score the student's answer as follows: Score: 85 |
| (b) | ... in one day, Faraday sold 13 + 84 = 97 sunflowers. In 3 days, Faraday sold 97 sunflowers/day. 3 days = 291 sunflowers. Therefore the answer is 291. (This answer is actually correct.) | The student incorrectly calculates the total number of sunflowers sold in 3 days. They add the total number of sunflowers sold in one day (97) to itself for 3 days, resulting in 291 sunflowers. Correct Calculation: Total sunflowers sold in 3 days = 97 sunflowers/day 3 days = 291 sunflowers. Therefore, the student's answer is incorrect due to the calculation mistake Therefore the score is: 85 |

