# OpenReview forum: "Demonstration Distillation for Efficient In-Context Learning"
_ICLR.cc/2024/Conference — Submitted to ICLR 2024_

### Official Review · Reviewer_rEVh · 2023-10-27

**Soundness:** 2 fair
**Presentation:** 2 fair
**Contribution:** 2 fair
**Rating:** 3
**Confidence:** 4

**Summary:**

The authors propose Distillist-Generalist-Specialist (DGS), an approach to reduce the input size of in-context learning examples without sacrificing accuracy.
DGS consecutively applies three modules, each of which is an LLM with a particular prompt:
The 'Distillist' receives a list of input-output pairs which it then attempts to 'distill' into shorter versions, without losing any information.
The 'Generalist' checks if each of the proposed distillations meet the following two general criteria: (1) the input message is only a question with no additional content or context, (2) the output response contains all information necessary to derive the final answer.
If the proposed distillation do not pass the test of the generalist, the distillist is re-queried.
Finally, the 'Specialist' consists of two steps: first, an LLM is used to try to answer the distilled question. Then a second LLM takes the distilled question and LLM prediction as input and predicts a 'score' for the distillation.
The LLM is prompted to produce a score that reflects the 'accuracy', 'relevance', and 'coherence' of the distillation – the prompt contains general instructions as to how the score should be produced.
If the LLM produces a score that is sufficiently high, the distillation is accepted.


The authors evaluate their approach on the GSM8K, BoolQ, and MultiRC tasks with ChatGPT, ChatGLM, and AutoCompressor.
They also compare against the AutoCompressor model as a baseline.

**Strengths:**

I agree with the authors that context size can be a limiting factor for In-Context Learning, e.g. when inputs exceed maximum context size or when the cost of using these models depends on the input size.
Compressing the size of inputs is an interesting way of solving these issues.
DGS presents an interesting and novel 'LLM-first' approach to implementing this solution.

**Weaknesses:**

Unfortunately, I believe there are a variety of issues with the draft in its current form that I would like to see the authors address before I can recommend its acceptance.

A) One concern with the paper is the lack of a zero-shot baseline.
For example, for BoolQ, the authors report a 3-shot accuracy of about 60% in Table 3.
This seems to be about the same as the _zero_-shot accuracy for GPT-3 (which should match or be outperformed by ChatGPT used by the authors) published here: https://paperswithcode.com/sota/question-answering-on-boolq?tag_filter=188.
**If there are no benefits to including in-context examples in the first place, then it is no surprise that there is no performance hit from compressing input examples with any method, including DGS.**
The zero-shot baseline can be seen as an 'infinite compression baseline': it compresses inputs of any size into zero tokens.
I would like the authors to demonstrate that DGS outperforms this zero-shot baseline.

B) Relatedly, Table 1 shows that there are _no_ improvements as the number of in-context examples is increased for GSM8K. In fact, an increase in the number of in-context learning (ICL) examples leads to decreased performance. On the one hand, this observation supports my worries that a zero-shot baseline would outperform DGS.
On the other hand, this makes me sceptical that DGS is actually a method that 'ensure generalizability across diverse query types'.
For standard ICL tasks, I would expect the performance to improve (and eventually stagnate) as more examples are provided in-context.

If there are no errors in this evaluation, I believe it is possible the authors have chosen tasks that do not really fit with common expectations for in-context learning.
If this is the case, I would ask the authors to provide evidence their method works on ICL tasks where more examples improve performance, either by repeating the experiments of Table 1 for  BoolQ and MultiRC or by evaluating on additional tasks for which this is definitely the case,  e.g. SST-2, Hate Speech, AG News, Subjectivity, RTE, MNLI, Financial Phrasebook).

C) I would suggest the authors make even more clear that their method only applies to targets task where the size of individual inputs is very large. (Alternatively, they can also demonstrate it works for other common ICL tasks such as SST-2, Hate Speech, AG News, Subjectivity, RTE, MNLI, or Financial Phrasebook.)

D) What is the uncertainty (e.g. boostrap confidence intervals) on the values in Table 1, 2, and 3? You claim that DGS "uniformly" improves performance. However, often, the gap is only 0.1%, and I am sceptical these results are significant.

For example, if we assume that, in Table 1, for GSM8K, performance does not actually decrease as we add more examples but, rather, it stays the same, this would mean the standard deviation is about ~1 percent.

You should be able to compute bootstrap confidence intervals from the numbers you already have, without running any new experiments.


E) Why are there no results for AutoCompressor on GSM8K in Table 2?


F) The authors should provide additional ablations/baselines.  The appendix contains an ablation for DGS without the generalist, however this does not provide accuracy  / compression ratio numbers.
I would be interested to see the performance (in terms of accuracy/compression ratio) without the generalist, as well as without the specialist.
Further, I would be interested to see the performance of the following simple baseline:
For each QA pair separately, you ask ChatGPT to 'Rewrite this input in the most concise way without losing any information that is necessary to arrive at the final answer' (and variations thereof).

G) Your introduction of the scoring system for the generalist is misleading. In the main body of the text you write that you will regenerate distillations if there are more than 10 points deducted. However, figure 7 reveals that a 10 point deduction is incurred from a single violation already. Why don't you just write that you regenerate if the generalist detects a single violation or more?


H) Sometimes the paper is a bit strong in claiming things like they are 'facilitating the reasoning capabilities of LLMs under the fixed constraint of context limit'.
For this to hold true, I would say they would need to show that, without DGS, prediction quality degrades.
E.g. only with DGS can I include sufficient examples to get reasonable predictions.
However, in fact, they just show they _reduce_ input size without taking a hit in accuracy. They do not show the opposite: _improving_ accuracy by being able to include a larger effective number of examples.


I) "Although successful, such fine-tuning processes necessitate substantial computational resources, often requiring hundreds of high-end GPUs, thereby increasing financial constraints for many researchers" --> To be fair, this only has to be performed once per model, whereas your approach has to be run once per task. While I think DGS is a valid idea, I think this argument against scalable positional embeddings does not hold up.

**Questions:**

J ) Have you tried going through QA pairs individually instead of asking the models to distill them in batches? It seems like this would possibly be less error-prone.

K) In the prompt for the specialist (figure 8), why do you not include the ground truth answer or the LLMs prediction for the original, long input? Surely, this would make it much easier to judge the usefulness of the distillation.

L) If I understand correctly, the distillist distils both questions and answers. Yet, the specialist does not check if the distilled answer makes sense – it only sees an 'initial response' produced from an LLM (based on the distilled question). Why is there no separate checking of the distilled answers (in addition to the general checking of the distillist)?

M) "If the computed score exceeds 90, it indicates that the distilled demonstrations are well-suited for the specific question. In this case, the demonstrations, having been evaluated by both the Generalist and Specialist, proceed to the Distillist for further refinement" --> Why, if the distilled demonstrations are 'well-suited' are they passed to the distillist again? I thought things would be done by now? Do you apply DGS iteratively or does the distillist do something else when it is called again? When do you break out of this loop?

N) "Demonstrations exceeding 2000 tokens are partitioned into two segments, each of which is distilled independently before concatenation. Additionally, since the input question participates in the distillation process, a question is randomly selected from the training set before distillation starts."  --> Why do you not use the question that actually belongs to the demonstration? The distillist prompt mentions "several User-Response pairs" but you just call it with a single input then?

O) "DGS proves consistently effective, regardless of how the demonstration and question are selected, as will be demonstrated in subsequent experiments" --> Where do you show this?

P) "Such an emergent feature fundamentally" --> Why is this an 'emergent' feature? What do you mean by this?
I do not think you have provided sufficient evident and discussion to claim this is an 'emergent' feature.

---

> ### Author Response · Authors · 2023-11-20
>
> We sincerely thank the reviewer for the constructive and detailed feedback. We respond to some of the reviewer’s concerns and questions below.
> ## Weakness A and B: The lack of a zero-shot baseline, and why there are no improvements as the number of demonstrations increases from 8 to 16 on GSM8K
> We fully understand the reviewer's question on why the performance on GSM8K degrades when changing from 8-shot demonstrations to 16-shot demonstrations. We have conducted experiments on GSM8K using 0-shot, 4-shot, 8-shot, and 16-shot randomly sampled demonstrations without any distillation (each of the results is averaged over three repeated runs, and all the experiments use ChatGPT-0925):
>   1. 0-shot: 72.5
>   2. 4-shot: 75.5
>   3. 8-shot: 77.2
>   4. 16-shot: 77.1
>
> The results above show that GSM8K is not a kind of task whose 0-shot evaluation outperforms few-shot, and it actually fits the general expectation of ICL. The interesting phenomenon is that the average performance of 16-shot demonstrations is still slightly outperformed by that of 8-shot. We reason that it is because 16-shot demonstrations are too long in context length (typically over 2000 tokens) and thus lead to higher difficulty for LLM to fully understand the context and relate to the test question, eventually failing to further improve the ICL performance. What our DGS does is just to alleviate such a long-context problem. As is demonstrated in Table 1, for 16-shot random demonstrations containing 2014 tokens and 8-shot longest demonstrations containing 3020 tokens, by distilling the overlong demonstrations to a reasonable length of around 1000 tokens so that the LLM can more easily understand, DGS both improves the original performance by 0.3% and 1% respectively.
>
> ## Question J: Why not try distilling QA pairs individually?
> We agree with the reviewer that distilling QA pairs individually instead of using a batch mode will possibly be less error-prone. But there are two noteworthy points. First, going through QA individually scales the latency cost by multiple API requests, and also increases the token cost since each pair of QA shares a large proportion of similar tokens in their input prompts(e.g. Distillation criteria provided for the LLM) and generated content from the LLM. Second, it is shown empirically that the batch mode we employ now already has a low error frequency, and is generally robust enough for the experimented demonstration distillation tasks.
>
> ## Question K: Why not provide the ground truth answer to the target question for the generalist?
> In our experiments, a target question is randomly sampled from the training set and does have a ground truth answer. We agree with the reviewer that including the ground truth answer to the target question will make it easier to judge the usefulness of the distillation, especially for the specialist. However, we choose not to include it out of the consideration of simulating real-world use cases. Ideally, a user may provide a target question along with the demonstrations for DGS, hoping that the target question can positively guide and check the distillation output. Since it is not always the case that the user knows the ground truth answer to the target question he provides, we think it is reasonable to assume that the ground truth answer is not given to DGS, and DGS should do all the distillation and checking only based on the target question itself.
>
> ## Question M: Clarification on the termination criteria of DGS
> For the termination criterion of our iterative DGS loop, we would like to state it more clearly as follows. Generally speaking, we hope the distilled demonstrations are as concise as possible under the condition of effectiveness (correctness). The distillist and generalist are mostly dedicated to the correctness of distillation and do not pay much attention to conciseness. Thus we hope the specialist can somehow not only check the effectiveness of the distilled demonstrations with regard to the user-provided question but also promote the degree of conciseness. We implement this idea by designing the special termination criteria in the specialist: if the computed score given by the specialist exceeds 90, it means the current distilled demonstrations can still serve the purpose of helping the LLM to answer the final question well, which indicates room for further distillation that continues to condense the currently eligible demonstrations to make it more concise. However, if the computed score falls below 90, then we think the demonstrations are now overly distilled, and cannot guide the solution of the final answer well enough. Therefore, the distillation process is terminated at this point, breaking out the DGS loop with the distilled demonstrations of the last round (i.e. The demonstrations that are most concise before being overly distilled).

---

> ### Author Response · Authors · 2023-11-20
>
> We sincerely thank the reviewer for the constructive feedback. This is the second part of our response to the reviewer’s concerns and questions below.
> ## Question N: Why not use one of the questions that actually belongs to the demonstrations as the target question?
> We would also like to clarify the components of the input to distillist. The distillist mainly receives 2 things. The first is the "User-Response" pairs, which are namely the demonstrations to be distilled. The second is a randomly sampled question from the training set that serves the purpose of checking if the distilled demonstrations are still eligible by the specialist. It is noteworthy that the randomly sampled question cannot in the meanwhile be part of the demonstrations to be distilled. Otherwise, it won't be able to be used for the specialist's checking procedure where the distilled demonstrations are prepended to the randomly sampled question for inference and evaluation.
> ## Question O: How to show the performance of DGS consistent among different demonstrations and questions?
> We fully understand the reviewer's question on how we demonstrate the robustness of DGS against the randomness posed by the varieties of demonstrations to be distilled. To address this, we have conducted a new experiment that randomly samples different 8-shot demonstrations for distillation, and calculate the average metric values including performance improvement and distillation ratio. The results on GSM8K are shown as follows:
> | # | Model         | # Token Original | # Token DGS | Ratio | Acc Original | Acc DGS |
> |---|---------------|------------------|-------------|-------|--------------|---------|
> | 1 | ChatGPT-0613  | 990              | 616         | *1.61 | 80.7         | 81.8    |
> | 2 | ChatGPT-0925  | 1096             | 490         | *2.24 | 76.1         | 77.1    |
> | 3 | ChatGPT-0925  | 1021             | 514         | *1.99 | 78.3         | 78.6    |
>
> The average distillation ratio is ×1.95, and the average performance improvement is 0.8%. These two metrics are both consistent among repeated runs, thus showing that our DGS is robust enough against randomness in selected demonstrations.
> ## Question P: What does "an emergent feature" mean?
> We thank the reviewer for pointing out this statement which is not rigorous enough in the submission. By saying "emergent", we intuitively posit that the ability to "Effectively Incorporate" is more likely to be present in larger language models like ChatGPT compared with smaller models, since such an ability requires understanding and rephrasing of multi-hop complex reasoning statements. However, we did not compare with more language models of different sizes or conduct a quantitative analysis on this problem, so such a statement is indeed not rigorous enough. We apologize for our oversight and have removed this statement from the revised paper.

---

> ### Comment · Reviewer_rEVh · 2023-11-21
> **Author Response**
>
> I thank the authors for their response to some of my concerns.
>
> While I thank the authors for the effort they put into the rebuttal, I am, at this point, still left with too many doubts regarding the proposed method to change my score.
>
> Most importantly, the authors have not not been able to show clearly that DGS consistently improves over a zero-shot baseline for the scenarios studied.
> I am also left with concerns about the reliability of the results and methodological choices about the design of the method.
>
>
> > Concerns A & B
>
> Could you address my concerns about the BoolQ dataset?
>
> Thanks for providing zero-shot numbers for ChatGPT. Why do the 4/8/16 shot numbers not match the ones in Table 1?
>
> I think it's necessary for you to include a dependable zero-shot baselines for all datasets in your paper.
>
> > Concerns C, D, E, F, G, H, I
>
> It would have been great if you had addressed these concerns.
>
> > Concerns J
>
> Thanks for the clarification on this!
>
> > Concern K
>
> I actually think it is somewhat unreasonable that the ground-truth answer (i.e. the label) is not known in an in-context learning application. In fact, your approach also relies on this at other points of the algorithm. Therefore, I think you should maybe consider adding the ground truth answer to the prompt of the specialist.
>
> > Concern M
>
> Thanks for your clarification. It would be great to see this clarified in an updated version of the draft, if you think this would be useful to other readers as well.
>
> > Concern N
>
> "The second is a randomly sampled question from the training set that serves the purpose of checking if the distilled demonstrations are still eligible by the specialist" -What do you mean by this? Could you elaborate the purpose of this randomly selected question?
>
> Additionally, my concern N was also directed at how the 'splitting' of longer demonstrations works.
>
> > Question O
>
> What is the difference between the two rows with ChatGPT-0925? Are these not averages over the entire dataset?
>
> > Question P
>
> Thanks!

---

### Official Review · Reviewer_AeYf · 2023-10-28

**Soundness:** 3 good
**Presentation:** 2 fair
**Contribution:** 3 good
**Rating:** 5
**Confidence:** 4

**Summary:**

ICL becomes popular among LLM users. However, such context burdens on the computational overheads. This work proposes a DGS framework to distill the in-context demonstrations (a so-called novel paradigm trying to take advantage of the natural language redundancy to propose better natural language prompts). Specifically, such framework operates on the input tokens, through token selection and overall content rephrasing to distill the contexts, building upon three expert LLM agents (ChatGPT), one for distillation, one for punishment, and the other for evaluation. The whole pipeline looks interesting and effective under three NLP tasks (GSM8K - mathematical reasoning, BoolQ - binary classification, and MultiRC - Reading Comprehension) for three large language models (ChatGPT, ChatGLM, and AutoCompressor - two of them are very aligned RLHF large LMs).

**Strengths:**

* The Agent pipeline looks interesting, and the final result looks promising in distilling the demonstrations in both compression ratio and performance improvements. This demonstrates their practical value, even under such simple heuristic-driven pipeline.

* I am also enjoying their analysis part on "how distilled demonstrations perform better". It looks very intuitive as natural language redundancy. Some contents are repeating the previous semantics. In addition, the power of ChatGPT in locating relevant knowledge also looks interesting.

**Weaknesses:**

* This work has demonstrates its value at those aligned or large LMs. I am just curious about whether such technique could generalize to smaller LMs, such as GPT-2, or others, since some generalization behaviors might **not** be the same.

* For your experimental setup, I have noticed that you test on **one** randomly sampled ICL prompt per task. I am just curious about the robustness of your utility in handling many other ICL prompts. That is to say, it would be better for showing some average results (performance, and compression ratio).

* It will definitely be better to include any kinds of **efficiency** discussions. If we aim at deploying your technique, how it costs?

I am happy to raise the score if there are any improvements.

**Questions:**

See Weaknesses, especially the point #2 and #3.

---

> ### Author Response · Authors · 2023-11-20
>
> We sincerely thank the reviewer for the constructive feedback. We appreciate that the reviewer recognizes our agent pipeline as effective under its simplicity. We also appreciate the reviewer's focus on our analysis of how distilled demonstrations perform better. We respond to the reviewer’s concerns and questions below.
>
> ## Weakness 2: The robustness in handling different ICL prompts?
> We fully understand the reviewer's concern about the robustness of DGS given the randomness of sampled demonstrations to be distilled. To address this, we have conducted experiments that sample different random demonstrations for distillation and calculated average results including performance improvement and distillation ratio. The results on GSM8K are shown as follows:
> | # | Model         | # Token Original | # Token DGS | Ratio | Acc Original | Acc DGS |
> |---|---------------|------------------|-------------|-------|--------------|---------|
> | 1 | ChatGPT-0613  | 990              | 616         | *1.61 | 80.7         | 81.8    |
> | 2 | ChatGPT-0925  | 1096             | 490         | *2.24 | 76.1         | 77.1    |
> | 3 | ChatGPT-0925  | 1021             | 514         | *1.99 | 78.3         | 78.6    |
>
>   The average distillation ratio is 1.95, and the average performance improvement is 0.8%. These two results are both consistent among the repeated experiments shown above.
>
> ## Weakness 3: Any efficiency discussions?
> We also thank the reviewer for pointing out the necessity of efficiency discussion. We would like to discuss the problem of cost and efficiency in two aspects.
> 1. First, we have conducted a new experiment to show that demonstration distillation for a given task by DGS can maintain the average cost within a reasonable budget. Specifically, we randomly select 8-shot demonstrations from GSM8K and distill them using the DGS method. During distillation, we record the number of times calling the distillist, generalist, and specialist respectively until the distillation is completed. We also record the total number of input and output tokens consumed in the whole procedure. The experiment is repeated 10 times and the final results are reported below. We have also updated the paper with this experiment in Appendix C.
> | No. | \# Distillist calls | \# Generalist calls | \# Specialist calls | \# Input tokens | \# Output tokens |
> |-----|------------|------------|------------|--------|--------|
> | 1   | 7          | 6          | 2          | 18011  | 5557   |
> | 2   | 3          | 2          | 2          | 8242   | 2630   |
> | 3   | 5          | 4          | 3          | 11085  | 3472   |
> | 4   | 3          | 2          | 2          | 7954   | 2377   |
> | 5   | 5          | 4          | 3          | 13368  | 4390   |
> | 6   | 2          | 1          | 1          | 5957   | 1256   |
> | 7   | 7          | 6          | 4          | 19635  | 5909   |
> | 8   | 5          | 4          | 1          | 11489  | 3160   |
> | 9   | 5          | 4          | 1          | 12485  | 4747   |
> | 10  | 3          | 2          | 1          | 8884   | 3356   |
> | **Mean** | **4.5 (±1.7)** | **3.5 (±1.7)** | **2 (±1.0)** | **11711 (±4391)** | **3685 (±1462)** |
>
>   From the data above we can estimate the average API usage cost. If we use OpenAI GPT-3.5-turbo-1106 API, then the average cost of the whole DGS procedure for a given task is (11711\*0.001 + 3685.4\*0.002) / 1000 = \$0.019. Such a cost is reasonable since our DGS is not a test-time method and only needs to be applied once to a given task.
>
>   2. Second, it is natural to compare DGS with other finetuning-based methods aimed at extending the context limit for LLMs, under the discussion of reducing costs. We would like to point out that our DGS method is an API-only procedure that does not require any GPU resource or computational cost aside from a reasonable cost of API usage. This makes our method more user-friendly and easy to use especially for practitioners who do not have much experience in LLM deployment and training, compared with the potential cost of data engineering and training intricacies involved in model finetuning.

---

### Official Review · Reviewer_VMEd · 2023-10-28

**Soundness:** 3 good
**Presentation:** 3 good
**Contribution:** 3 good
**Rating:** 3
**Confidence:** 4

**Summary:**

This paper introduces a strategy called Distillist-Generalist-Specialist to distill demonstrations in In-context Learning. Although the method is straightforward, it offers a practical approach to addressing the challenges presented by long-text demonstrations. There are some questions about this paper and would like the authors to provide further clarification.

**Strengths:**

1. This paper proposes a simple strategy to compress the content of demonstrations, resulting in shorter textual lengths. This is particularly helpful for long-text demonstrations.

2. The authors provide a clear demonstration of the approach and conduct numerous case studies, allowing for a quick understanding of how this method is implemented.

**Weaknesses:**

1. The innovation in this paper is lacking. Using large models to compress content and address the issue of excessively long text is a technique that has been applied for quite some time. For instance, techniques like summarization in langchain-github have been used to avoid exceeding maximum length limits.

2. The description of the key aspects of this method is not precise enough, and I couldn't find a clear definition of "Punishment Score."

3. The applicability of the method is not adequately clarified. For example, the mathematical examples showcased by the authors may be easily compressed appropriately for ChatGPT. However, for tasks involving causal reasoning, sarcasm, or emotional inference, using large models to compress text length may likely result in the loss of crucial information, which is not desirable.

4. The lack of evidence for improvement in accuracy makes the explanations seem more like speculation. The compression that removes incorrect information appears similar to a CoT process, but is the test input compressed? If it is, it becomes difficult to verify if the improvement is due to changes in demonstrations because the test input questions have also changed.

**Questions:**

1. Techniques like summarization in langchain-github been used to avoid exceeding maximum length limits, does this paper's contribution only lie in proposing a few prompts for compressing demonstrations?

2. Could you provide a clear definition of "Punishment Score"? I couldn't find the mathematical definition of this term. Is this value just evaluated by ChatGPT? How is its reliability ensured?

3. How about the applicability of the method? The mathematical examples showcased by the authors may be easily compressed appropriately for ChatGPT. However, for tasks involving causal reasoning, sarcasm, or emotional inference, using large models to compress text length may likely result in the loss of crucial information.

4. The rewriting seems like a CoT (Compression of Transformations) process that changes the way reasoning is done. It's reasonable to say that a more stright question has a higher accuracy for the model. However, if the final test questions are compressed, it becomes difficult to verify whether the improvement is due to changes in demonstrations. This is because the input questions have changed. For example, if a mathematical problem is rephrased by ChatGPT to make it clearer, it becomes easier to answer correctly. However, this doesn't prove that the improvement is due to the author's method or its connection to demonstrations. It may simply be the result of the question being rephrased, which is different from the claim made by the author regarding their contribution.

---

> ### Author Response · Authors · 2023-11-20
>
> We sincerely thank the reviewer for the constructive feedback. We respond to the reviewer’s concerns and questions below.
> ## Weakness 1 (Question 1): Difference between DGS and langchain summarization?
> We fully understand the reviewer's concern about our paper's novelty compared with summarization techniques used in langchain. We think there exist fundamental differences between our DGS and langchain summarization, mainly in designing motivation and scope of application. First, our DGS method is specifically designed for "distilling" contexts instead of "compressing" contexts. The key difference between distillation and compression is that distillation is always performed and checked with regard to the specific question provided by the user, which is especially highlighted by the specialist. To be specific, the specialist checks if the distilled demonstrations presented by the distillist are over-distilled by prepending these demonstrations to the user-given question and testing if the LLM's response to the given question is still satisfactory. On the contrary, the techniques in langchain only focus on the simple and general task of summarizing a given document, without an effective checking procedure that is specially optimized for a few-shot demonstration setting or taking into consideration the effect of distilled demonstrations with respect to the user-provided question. Moreover, table 4 in Appendix B also shows some examples of key information loss in distilled few-shot demonstrations when the checking procedure is removed, which substantiates the importance of the completeness of our DGS structure.“”
>
> ## Weakness 2 (Question 2): The definition of Punishment Score?
> We provide a definition of the Punishment Score in the prompt given to the generalist in Figure 7. Specifically, the punishment score is imposed by the generalist if the following criterion is not met: there exists one distilled demonstration whose answer part uses key information that is not provided in its question part but still derives the final answer, thus indicating over-distillation. Each ineligible demonstration under this criterion will increase the Punishment Score by 10 points.
>
> ## Weakness 4 (Question 4): Whether the test question is compressed?
> We would also like to point out that during the DGS process, the test question will not be involved and thus won't be distilled. The user-provided question used in DGS is a randomly sampled one from the same distribution as the demonstrations to be distilled (i.e. The training set). The whole distillation process is completed before the distilled demonstrations are prepended to the test question for inference and evaluation.

---

### Official Review · Reviewer_u1NF · 2023-10-31

**Soundness:** 2 fair
**Presentation:** 4 excellent
**Contribution:** 2 fair
**Rating:** 3
**Confidence:** 4

**Summary:**

The paper tries to reduce computational burdens and financial costs of LLMs by demonstration distillation. The motivation is that in ICL, context or demonstrations take up many tokens in the prompt, but a large number of tokens may not be feasible for current LLMs. So, this paper proposes a method to remove redundant tokens in prompts and try to keep the ICL performance. The proposed method DGS contains three agents, i.e., Distillist, Generalist and Specialist. They are responsible for different parts of the task, and complete the whole distillation process in a cooperative and iterative manner. The proposed DGS is evaluated on multiple benchmarks, and good results are obtained.

**Strengths:**

This paper is well-motivated, since reducing the prompt length is necessary in many real-world scenarios. The presentation in this paper is good and the idea behind it is illustrated clearly with explanations and examples. The design method is reasonable to some extent, since many aspects when doing demonstration distillation are considered in DGS, such as retaining essential elements, controlling quality, accuracy, relevance, and so on.

**Weaknesses:**

It seems that many costs are introduced by the proposed method, but the proposed method aims at reducing costs at the beginning. In order to do distillation, three agents, with ChatGPT behind them, are called iteratively, which must cause many computational burdens and financial costs. So, taking the costs in this distillation process into account, will the DGS reach the goal of reducing costs? For the cases where the context already exceeds the length limit of the LLM, how does this compression take effect?

**Questions:**

Is the randomness from LLMs considered in experiments? Using a LLM without randomness or setting the temperature to 0 is a more appropriate way for reproducible experiments.

---

> ### Author Response · Authors · 2023-11-20
>
> We sincerely thank the reviewer for the constructive feedback. We appreciate that the reviewer regards our work as well-motivated and our method design as reasonable. We respond to the reviewer’s concerns and questions below.
> ## Weakness 1: Can DGS reach the goal of reducing costs?
> We fully understand the reviewer's concern about DGS's ability to reduce costs. We would like to address this concern in two aspects.
>   1. First, we have conducted a new experiment to show that demonstration distillation for a given task by DGS can maintain the average cost within a reasonable budget. Specifically, we randomly select 8-shot demonstrations from GSM8K and distill them using the DGS method. During distillation, we record the number of times calling the distillist, generalist, and specialist respectively until the distillation is completed. We also record the total number of input and output tokens consumed in the whole procedure. The experiment is repeated 10 times and the final results are reported below. We have also updated the paper with this experiment in Appendix C.
> | No. | \# Distillist calls | \# Generalist calls | \# Specialist calls | \# Input tokens | \# Output tokens |
> |-----|------------|------------|------------|--------|--------|
> | 1   | 7          | 6          | 2          | 18011  | 5557   |
> | 2   | 3          | 2          | 2          | 8242   | 2630   |
> | 3   | 5          | 4          | 3          | 11085  | 3472   |
> | 4   | 3          | 2          | 2          | 7954   | 2377   |
> | 5   | 5          | 4          | 3          | 13368  | 4390   |
> | 6   | 2          | 1          | 1          | 5957   | 1256   |
> | 7   | 7          | 6          | 4          | 19635  | 5909   |
> | 8   | 5          | 4          | 1          | 11489  | 3160   |
> | 9   | 5          | 4          | 1          | 12485  | 4747   |
> | 10  | 3          | 2          | 1          | 8884   | 3356   |
> | **Mean** | **4.5 (±1.7)** | **3.5 (±1.7)** | **2 (±1.0)** | **11711 (±4391)** | **3685 (±1462)** |
>
>   From the data above we can estimate the average API usage cost. If we use OpenAI GPT-3.5-turbo-1106 API, then the average cost of the whole DGS procedure for a given task is (11711\*0.001 + 3685.4\*0.002) / 1000 = \$0.019. Such a cost is reasonable since our DGS is not a test-time method and only needs to be applied once to a given task.
>
>   2. Second, it is natural to compare DGS with other finetuning-based methods aimed at extending the context limit for LLMs, under the discussion of reducing costs. We would like to point out that our DGS method is an API-only procedure that does not require any GPU resource or computational cost aside from a reasonable cost of API usage. This makes our method more user-friendly and easy to use especially for practitioners who do not have much experience in LLM deployment and training, compared with the potential cost of data engineering and training intricacies involved in model finetuning.
>
> ## Weakness 2: How does DGS take effect when the context already exceeds the length limit of the LLM?
> For the second concern that the original context may exceed the LLM's context limit, we actually address it in the paper by first partitioning demonstrations that exceed 2000 tokens into two segments and distilling each segment respectively, and finally concatenating together the distilled contents.
>
> ## Question 1: How is randomness from LLMs considered in experiments?
> Since our focus is on developing a more convenient-to-use and API-only method, we add the testing of the robustness of DGS against the inherent randomness of ChatGPT by repeating the following experiment on GSM8K 3 times: first randomly select 8-shot demonstrations and distill them using DGS, and next test the performance of the distilled demonstrations compared with original ones. The supplementary results are as follows:
> | # | Model         | # Token Original | # Token DGS | Ratio | Acc Original | Acc DGS |
> |---|---------------|------------------|-------------|-------|--------------|---------|
> | 1 | ChatGPT-0613  | 990              | 616         | *1.61 | 80.7         | 81.8    |
> | 2 | ChatGPT-0925  | 1096             | 490         | *2.24 | 76.1         | 77.1    |
> | 3 | ChatGPT-0925  | 1021             | 514         | *1.99 | 78.3         | 78.6    |
>
>   The supplementary experiments above show that our results are robust and reproducible given the inherent randomness of ChatGPT.

---

### Official Review · Reviewer_Ddz4 · 2023-11-01

**Soundness:** 2 fair
**Presentation:** 3 good
**Contribution:** 2 fair
**Rating:** 3
**Confidence:** 5

**Summary:**

The paper proposes an approach to compress demonstrations that can be used in an in-context learning setup. The proposed approach is termed as Distillist-Generalist-Specialist (DGS) which as the name suggests consists of three LLM-powered agents that help with the compression process. This approach shows similar performance to uncompressed demonstrations while significantly reducing the number of input tokens. The proposed approach is designed to help reduce computational and financial costs that arise from using lengthy demonstrations.

**Strengths:**

The paper is well written and is easily understandable. The proposed approach is simple and intuitive. Empirical experiments demonstrate the value of the proposed DGS approach by significantly reducing the number of input tokens and retaining overall performance.

**Weaknesses:**

1) The proposed DGS approach is powered using OpenAI's chatgpt. While chatgpt is a very popular and capable model, nonetheless it is a closed-source model. The authors should have investigated a few (1-3) open-source LLMs that could power the three LLM agents instead and check how the performance changes.

2) The proposed approach is motivated to help reduce computational and financial costs that are associated with using longer demonstrations in an ICL framework. However, using models like chatgpt to power the proposed approach can also end up being expensive to operate. The authors should have designed and included a small experiment describing the potential financial relief the proposed approach offers.

3) The proposed approach is only compared against AutoCompressor method despite including two other demonstration compression methods in the related work section.

**Questions:**

1) Can you test your approach using 1-2 open-source models and evaluate how that affects performance?
2) Can you include performance results using the demonstration compressions methods that were listed in the related work section?
3) Compare your approach to other demonstration selection approaches that specifically look for valid but short demonstrations that can be used in an ICL framework. Look at the following papers: a) https://aclanthology.org/2023.findings-acl.273.pdf; b) https://arxiv.org/abs/2211.04486; c) https://arxiv.org/abs/2310.10707; d) https://arxiv.org/abs/2302.05698; e) https://arxiv.org/abs/2104.08786

---

> ### Author Response · Authors · 2023-11-20
>
> We sincerely thank the reviewer for the constructive feedback. We appreciate that the reviewer regards our work as well-written and our approach as simple and intuitive. We respond to the reviewer’s concerns and questions below.
> ## Weakness 1 (Question 1): Why not test DGS's performance with open-source LLMs?
> We fully agree with the reviewer that it is beneficial to test DGS's performance with open-source LLMs so as to gain a more comprehensive picture of DGS's applicability. However, we would like to point out that the black-box property of DGS is also one of the most important considerations as we design its implementation. We specifically design DGS as an API-only procedure that does not require any GPU resource. This makes our method more convenient to use, especially for general practitioners who do not have much experience or abundant resources in LLM deployment. We deem such an API-only and user-friendly feature of DGS as one of its most valuable merits and therefore do not prioritize the testing with various open-source LLMs.
> ## Weakness 2: How much financial relief does DGS offer, compared with other finetuning-based approaches to extending context-limit?
> We fully understand the reviewer's concern about DGS's ability to reduce costs. We would like to address this concern in two aspects.
>   1. First, we have conducted a new experiment to show that demonstration distillation for a given task by DGS can maintain the average cost within a reasonable budget. Specifically, we randomly select 8-shot demonstrations from GSM8K and distill them using the DGS method. During distillation, we record the number of times calling the distillist, generalist, and specialist respectively until the distillation is completed. We also record the total number of input and output tokens consumed in the whole procedure. The experiment is repeated 10 times and the final results are reported below. We have also updated the paper with this experiment in Appendix C.
>
> | No. | \# Distillist calls | \# Generalist calls | \# Specialist calls | \# Input  tokens| # Output tokens |
> |-----|------------|------------|------------|--------|--------|
> | 1   | 7          | 6          | 2          | 18011  | 5557   |
> | 2   | 3          | 2          | 2          | 8242   | 2630   |
> | 3   | 5          | 4          | 3          | 11085  | 3472   |
> | 4   | 3          | 2          | 2          | 7954   | 2377   |
> | 5   | 5          | 4          | 3          | 13368  | 4390   |
> | 6   | 2          | 1          | 1          | 5957   | 1256   |
> | 7   | 7          | 6          | 4          | 19635  | 5909   |
> | 8   | 5          | 4          | 1          | 11489  | 3160   |
> | 9   | 5          | 4          | 1          | 12485  | 4747   |
> | 10  | 3          | 2          | 1          | 8884   | 3356   |
> | **Mean** | **4.5 (±1.7)** | **3.5 (±1.7)** | **2 (±1.0)** | **11711 (±4391)** | **3685 (±1462)** |
>
>   From the data above we can estimate the average API usage cost. If we use OpenAI GPT-3.5-turbo-1106 API, then the average cost of the whole DGS procedure for a given task is (11711\*0.001 + 3685.4\*0.002) \/ 1000 = \$0.019. Such a cost is reasonable since our DGS is not a test-time method and only needs to be applied once to a given task.
>
>   2. Second, it is natural to compare DGS with other finetuning-based methods aimed at extending the context limit for LLMs, under the discussion of reducing costs. We would like to point out again that our DGS method is an API-only procedure that does not require any GPU resource or computational cost aside from a reasonable cost of API usage. This makes our method more user-friendly and easy to use especially for practitioners who do not have much experience in LLM deployment and training, compared with the potential cost of data engineering and training intricacies involved in model finetuning.

---

> ### Author Response · Authors · 2023-11-20
>
> We sincerely thank the reviewer for the constructive feedback. This is the second part of our response to the reviewer’s concerns and questions below.
>
> ## Question 3: Why not compare DGS with other demonstration selection approaches that specifically look for valid but short demonstrations?
> We appreciate the reviewer's insightful suggestion to compare DGS with other demonstration selection approaches that specifically look for short but effective demonstrations. However, we think it may not be very necessary to conduct such a comparison since demonstration selection techniques are in essence different from demonstration distillation techniques. To be more specific, demonstration selection focuses on the "quality" of a small fraction of demonstrations in the training set from which the LLM can learn most. In contrast, demonstration distillation does not care much about the quality of the demonstrations it receives and can distill and condense whatever groups of input demonstrations, to an extent where distilled demonstrations are shorter than most of the original demonstrations in the whole training set. Even if the demonstrations chosen by special selection techniques are on par with the distilled demonstrations in length, the comparison between them is still not fair enough, since the distilled ones are randomly selected and there may exist an inherent quality gap between their original versions and the selected short but effective ones. Out of such considerations, we did not compare DGS with demonstration selection approaches in our submitted paper.
> ## Weakness 3 (Question 2): Why not compare DGS with demonstration compression methods other than AutoCompressor?
> We appreciate the reviewer's suggestion to compare our method with additional compression methods referenced in our related work. However, we respectfully contend that a comparison with the two specific works cited (https://arxiv.org/pdf/2304.08467.pdfandhttps://arxiv.org/pdf/2307.06945.pdf) may not be strictly necessary for our study.
>   - Regarding the paper "Learning to Compress Prompts with Gist Tokens," the authors have innovatively developed a technique for condensing short, instruction-like prompts into gist tokens. This approach, while effective for its intended purpose, seems less applicable to our goal of compressing longer demonstrations for in-context learning. Our focus is on distilling extensive contexts into shorter forms, which diverges from the gist token methodology.
>   - On the other hand, the paper "In-context Autoencoder for Context Compression in a Large Language Model" introduces the In-context Autoencoder (ICAE), a novel model for context compression. This method shares similarities with the AutoCompressor, as both approaches aim to compress contexts into vectors of a fixed shape using pretrained language models. However, it's important to note that our distillation method, while akin to these compression techniques in reducing context lengths, has a distinct objective. Our method is specifically designed to tailor demonstrations to optimally address target questions, as opposed to seeking a universal compression solution. In light of these differences, we believe that a comparison between our DGS and the AutoCompressor sufficiently highlights the unique aspects of our approach compared to existing compression methods, and therefore consider further comparison with ICAE as not so necessary.

---

> > ### Comment · Reviewer_Ddz4 · 2023-12-02
> > **Author response**
> >
> > I thank the authors for their response. However, I am still not convinced to change my original score. The proposed approach defeats the main motivation of reducing costs by distilling the random demonstrations. I still believe that careful selection and arrangement of demonstrations would still outperform and be cheaper than distilling demonstrations using the proposed approach. The proposed approach also lack comparison with other compression methods.

---

### Meta-Review · Area_Chair_t7fx · 2023-12-05

**Metareview:**

This paper tries to remove redundant tokens in the prompt while keeping the in context learning performance. For this the paper proposed the use of three agents, the Distillist, the Generalist and the Specialist.

There was a consensus among reviewers that this paper should be rejected. Key reasons were that only a single closed-source model was used, concerns that the learning transfer to different settings, and the novelty and efficiency of the approach.

**Justification For Why Not Higher Score:**

There was a consensus among reviewers that this paper should be rejected. Key reasons were that only a single closed-source model was used, concerns that the learning transfer to different settings, and the novelty and efficiency of the approach.

**Justification For Why Not Lower Score:**

N/A

---

### Decision · Program_Chairs · 2024-01-16

Reject